# Successful Treatment of a Rare Cholesterol Homeostasis Disorder Due to *CYP27A1* Gene Mutation with Chenodeoxycholic Acid Therapy

**DOI:** 10.3390/biomedicines11051430

**Published:** 2023-05-12

**Authors:** Petar Brlek, Luka Bulić, David Glavaš Weinberger, Jelena Bošnjak, Tomislav Pavlović, Svetlana Tomić, Zdravka Krivdić Dupan, Igor Borić, Dragan Primorac

**Affiliations:** 1St. Catherine Specialty Hospital, 10000 Zagreb, Croatia; 2Department of Health Studies, University of Split, 21000 Split, Croatia; 3School of Medicine, Josip Juraj Strossmayer University of Osijek, 31000 Osijek, Croatia; 4University Hospital Centre, 31000 Osijek, Croatia; 5Medical School, University of Split, 21000 Split, Croatia; 6Department of Biochemistry & Molecular Biology, The Pennsylvania State University, State College, PA 16802, USA; 7The Henry C. Lee College of Criminal Justice and Forensic Sciences, University of New Haven, West Haven, CT 06516, USA; 8Medical School REGIOMED, 96450 Coburg, Germany; 9Medical School, University of Rijeka, 51000 Rijeka, Croatia; 10Faculty of Dental Medicine and Health, Josip Juraj Strossmayer University of Osijek, 31000 Osijek, Croatia; 11Medical School, University of Mostar, 88000 Mostar, Bosnia and Herzegovina; 12National Forensic Sciences University, Gujarat 382007, India; 13University Hospital Centre Zagreb, 10000 Zagreb, Croatia

**Keywords:** ataxia, cerebrotendinous xanthomatosis, chenodeoxycholic acid, cholestanol, xanthoma, next-generation sequencing

## Abstract

Cerebrotendinous xanthomatosis (CTX) is a genetic disorder of the cholesterol metabolic pathway, most often associated with variants in the CYP27A1 gene. The dysregulation of cholesterol metabolism results in the accumulation of metabolites such as cholestanol, which has a predilection for neuronal tissue and tendons. The condition is treatable with chenodeoxycholic acid (CDCA), which halts the production of these metabolites. We present two adult brothers, without diagnosis, suffering from ataxia, general muscle weakness and cognitive deficits. Both brothers suffered from early onset cataracts, watery stools and thoracic kyphoscoliosis. Magnetic resonance imaging revealed hyperintense alterations in the central nervous system and intratendinous xanthomas in the Achilles tendons. A biochemical analysis showed elevated levels of cholestanol, lathosterol and 7-dehydrocholesterol. Their family history was negative for neurological and metabolic disorders. Genetic testing revealed a pathogenic *CYP27A1* variant (c.1184+1G>A) in both brothers, confirming the diagnosis. The patients were started on CDCA therapy and have shown significant improvement at their follow-up examinations. Early diagnosis and treatment initiation in CTX patients is of great importance, as the significant reversal of disease progression can be achieved. For this reason, clinical genetic testing is necessary when it comes to patients with an onset of cataracts, chronic diarrhea, and neurological symptoms in early childhood.

## 1. Introduction

Cerebrotendinous xanthomatosis (CTX) is an autosomal-recessive disease first described in 1937, caused by the dysregulation of the cholesterol metabolic pathway [1,2]. The disease presents a wide variety of systemic symptoms, but it predominantly affects the central and peripheral nervous systems. Neurological manifestations include abnormal behavioral patterns, dementia, convulsions, pyramidal paralysis and cerebellar, brainstem, spinal and peripheral nerve dysfunction. Other manifestations particularly affect the eyes, causing an early onset of cataracts, lungs, blood vessels and tendons. There are many reported cases of premature atherosclerosis in CTX patients, which often leads to life-threatening cardiovascular events at an earlier age. If left untreated, the disease progresses slowly and has a lethal outcome. The clinical diagnostic criteria are the finding of increased concentrations of bile alcohols in the serum and urine with normal or low serum cholesterol levels, while confirmation of the diagnosis is attained by genetic testing [3,4].

CTX is caused by homozygous or compound heterozygous variants in the *CYP27A1* gene. *CYP27A1* is situated on chromosome 2q35 and encodes the sterol 27-hydroxylase enzyme, a mitochondrial cytochrome P450 enzyme [4]. This enzyme catalyzes the first step in sterol intermediate oxidation and is necessary for bile acid synthesis. Dysfunction of the enzyme results in the activation of alternative sterol metabolic pathways and increased synthesis of 25-hydroxylated bile alcohols and cholestanol (Figure 1). These excess metabolites then form deposits in virtually all tissues, causing the aforementioned symptoms [5].

Genetic criteria for the confirmation of this disease are the finding of pathogenic variants in both alleles of the *CYP27A1* gene. Two variants of uncertain significance or one pathogenic variant and one variant of uncertain significance are not considered definitive proof. The testing can be performed either for the *CYP27A1* gene specifically, as a panel of multiple genes that includes *CYP27A1*, or as a clinical exome (CES) or whole genome sequencing (WGS) [6].

The main treatment option for CTX is chenodeoxycholic acid (CDCA), a bile acid that is naturally found in the body. Using the negative feedback mechanism, it downregulates the hepatic synthesis of CYP7A1, an enzyme that functions as the rate-limiting step in the synthesis of bile acids. This downregulation ultimately helps to reduce the accumulation of toxic bile acid intermediates in CTX patients. In patient studies, it was able to reduce cholestanol levels in the cerebral spinal fluid by three-fold and ultimately halt the progression of the disease [7].

This article presents two patients, brothers, diagnosed with CTX. It focuses on their clinical presentation and diagnostic results, as well as on their response to CDCA therapy.

## 2. Materials and Methods

### 2.1. Subjects

Two male patients, brothers, 40 (patient 1) and 36 (patient 2) years of age, suffer from extensive motor and cognitive deficits. After going through rehabilitation therapy for several years, they attended an examination at St. Catherine’s Specialty Hospital. No other family members shared this disorder.

### 2.2. Clinical Examination and Treatment

Patients were first physically examined, with an emphasis placed on their motor abilities and muscular tone. Their blood was drawn in order to perform basic biochemical analysis. Additionally, a sample of blood was sent for gas chromatography and mass spectrometry analysis to measure levels of cholesterol metabolites. A magnetic resonance imaging (MRI) examination of the central nervous system and lower leg was performed to discover potential changes or xanthomas. After confirming the diagnosis, the patients were placed on CDCA therapy (3 × 250 mg/day). The effectiveness of the treatment was monitored by evaluations of their clinical status, re-evaluations of the central nervous system using MRI and biochemical analysis of cholesterol and cholesterol metabolite levels. 

### 2.3. Genomic Testing

Genomic DNA was isolated from the peripheral blood of both patients and analyzed through a multigene panel using next-generation sequencing. The panel included the sequencing of 444 genes and covered a wide array of neurometabolic disorders. All target genes were sequenced to a minimum depth of 50×. Sequencing results were read in accordance with the reference genome (GRCh37). Promotors and other noncoding regions were excluded from the analysis. As described in our previous study, exon-level copy number (deletions and duplications) and other types of non-SNV variants were identified using validated bioinformatics algorithms [8].

### 2.4. Bioinformatics and Statistical Analysis

Since CYP27A1 variants have been associated with malignant diseases, we used the publicly available database cBioPortal (https://www.cbioportal.org/, accessed on 15 April 2023) for cancer genomics to analyze data on 19,823 patients/tumor samples and evaluate the implication of loss of CYP27A1 enzyme activity in carcinogenesis. Statistical analysis of the obtained data (overall survival time) was performed using the cBioPortal platform, with a significance level of *p* < 0.001.

## 3. Results

### 3.1. Phenotypes of the Patients

Although the phenotypes of our patients share most of the key elements, there are some characteristics specific to one, but not the other (Table 1).

Patient 1, a 40-year-old man, was born to non-consanguineous parents without complications. Early cognitive and motor development did not show pathological characteristics. At preschool age, he started showing clumsy behavior and would often fall, which led to fractures of his arm and clavicle. Throughout his childhood, he wore an orthosis for the treatment of kyphoscoliosis. At 16 years of age, he started having epileptic seizures, which are currently treated with anticonvulsant therapy. At the age of 31 he was diagnosed and surgically treated for bilateral early onset cataracts, and was later hospitalized for ataxia and frequent episodes of dizziness in the same year. Over the next two years, his disease progressed to paraparesis, and he was placed in continuous rehabilitation therapy. Following these findings, the patient was indicated for genetic testing. The patient has a healthy son, and, except for his brother, no other members of his family suffered from neurological disorders.

At his physical examination at St. Catherine’s Specialty Hospital, he required two crutches to support him while walking, which showed elements of ataxia. Other notable findings were severe thoracic kyphosis and general muscle weakness and hypotonia. A brain magnetic resonance imaging (MRI) showed hyperintense regions in the cerebellar white matter, caudate nucleus and periventricular area in the T2/FLAIR sequence. An MRI of the patient’s lower leg and ankles showed an intratendinous xanthoma, 17 × 44 mm in size, in the left Achilles tendon. Biochemical analysis of the patient’s blood showed elevated levels of cholesterol metabolites cholestanol, lathosterol and 7-dehydrocholesterol, but normal cholesterol levels.

Patient 2, a 36-year-old man, was born to non-consanguineous parents without complication. Early motor development was without pathological features. However, the patient had a notable delay in speech development. From birth, the patient experienced frequent watery stools. At the age of 7, he was surgically treated for bilateral juvenile cataracts. Psychological evaluations in his teenage years describe heavy cognitive underdevelopment. Motor symptoms developed at 25 years of age and manifested as ataxia. Given the finding, genetic testing was indicated. The patient was also placed in continuous rehabilitation therapy.

At his physical examination at St. Catherine’s Hospital, he also required two crutches to support his walking and showed elements of ataxia. Similarly to his brother, he had thoracic kyphosis, general muscle weakness and hypotonia. Unlike his brother, the patient had poor anal sphincter control and frequent diarrhea. The MRI examination of his central nervous system showed the same findings as in his brother, bilateral hyperintense changes in the cerebellar white matter, caudate nucleus and periventricular area in the T2/FLAIR sequence. An MRI examination of both of his ankles and lower legs showed two intratendinous xanthomas, 41 mm in size in the right Achilles tendon and 19 mm in the left Achilles tendon. Blood analysis also showed elevated levels of cholestanol, lathosterol and 7-dehydrocholesterol with normal cholesterol levels. 

### 3.2. Results of Genetic Testing

The neurometabolic disorder panel, using next-generation sequencing, discovered eight variants in the DNA of patient 1 and three variants in the DNA of patient 2 (Table 2).

The clinically significant finding of the panel is the c.1184+1G>A variant in *CYP27A1*. The variant is situated on the intron–exon border in the donor splice site. Donor splice site variants disrupt RNA splicing and can result in the loss of exons or the inclusion of introns. Either way, the product is an altered protein-coding sequence. The same variant was discovered in both brothers, excluding a de novo origin. Pathogenic variants in the *CYP27A1* gene have been associated in the literature with CTX. Given the pathogenic classification of the variant, this diagnosis was made for both brothers. The contribution of other variants to their condition cannot be inferred since the variants are classified as variants of uncertain significance.

### 3.3. The Effect of Chenodeoxycholic Acid (CDCA) Treatment

Both patients have had frequent follow-up examinations following the initiation of CDCA therapy, in which a notable improvement in clinical status was observed. Cholesterol metabolite levels were measured 3 months and 6 months after initiation of treatment. The results confirmed that the treatment was highly effective (Table 1). Their latest follow-up examination was ten months after starting treatment.

Patient 1 was able to stand without the assistance of crutches. He was able to stand up from a sitting position with minimal assistance. However, he still needed crutches to walk. Subjectively, he felt more stable and stronger in his lower extremities. The control MRI examination confirmed that there was no progression of lesion growth (Figure 2).

Patient 2 also described feeling stronger and more stable in his lower extremities. He was able to stand up from a sitting position without any assistance and could walk longer than before. His number of daily stools had significantly reduced and they were more solid in consistency. The control MRI examination confirmed that there was no progression of lesion growth (Figure 3).

The patients showed no side effects of the treatment throughout the follow-up period, including any hepatotoxic effects that were monitored by liver enzyme tests (Table 1). Six months after the initiation of the treatment, an increased level of total serum cholesterol and LDL were present in both patients, while HDL remained within range. Patient 1 had a total serum cholesterol of 6.00 mmol/L (H), while the HDL was 1.06 mmol/L (N) and the LDL was 4.20 mmol/L (H). Patient 2 had a total serum cholesterol level of 5.52 mmol/L (H), an HDL level of 1.05 mmol/L (N) and an LDL level of 3.78 mmol/L (H).

## 4. Discussion

We have identified two brothers with CTX caused by a homozygous pathogenic variant in the *CYP27A1* gene (c.1184+1G>A). Studies on skin fibroblasts have shown that the disruption of the splice site in intron 6 leads to the skipping of 89 nucleotides of exon 6 and introduces a premature termination codon [9]. The patients had a very typical clinical presentation for this condition, which consists of early onset cataracts, Achilles tendon deposits and neurological cognitive and motor dysfunction. Both brothers shared a similar progression of the disease, with cataracts and cognitive impairment occurring at an early age and motor impairment becoming dominant in their late 20s and early 30s. They were diagnosed with spinocerebellar ataxia and placed in rehabilitation therapy. There are multiple possibilities for the genetic analysis of patients with suspected CTX, from gene-targeted testing (single-gene testing and multigene panel) in patients with more indicative clinical findings, to exome and whole genome sequencing. Multigene genetic testing detects pathogenic variants of the *CYP27A1* gene in up to 99% of cases [6]. However, their attending physicians did not consider genetic testing necessary as they were not of the opinion that any results would impact their treatment or progression. Only after several years were they tested at St. Catherine’s Hospital, where this proved not to be the case. The progression of CTX can be halted very well using CDCA therapy. It is crucial to begin with a therapeutic regimen as early as possible, as there is a direct correlation between the time of initiation of therapy and the improvement in clinical status [10]. Had the patients been placed on this regimen earlier, they probably would have had a superior clinical outcome.

A notable characteristic of the clinical presentation of our patients is a relatively late onset of major neuromotor symptoms. While other symptoms of the disease were present from early childhood, such as juvenile cataracts, delay in cognitive development, epilepsy and diarrhea, general motor deficits of ataxia became severe in their late 20s and early 30s. Another case (Li ZR. et al.) describes a 38-year-old male patient suffering from CTX caused by a *CYP27A1* variant. This patient, however, suffered from ataxia and other severe motor deficits since the age of eight. Left untreated, the disease developed slowly over the course of 30 years, resulting in irreversible damage [11].

CTX has alternative treatments to CDCA, which also act as up-regulators of the cholesterol metabolic pathway. Examples of these are other bile acids, such as ursodeoxycholic acid (UDCA), cholic acid and taurocholic acid. Although the optimal treatment for the neurological and non-neurological symptoms of CTX is CDCA, cholic acid is also considered effective in treating non-neurological symptoms of the condition [10]. The advantage of CDCA over UDCA is that a smaller dose is needed to achieve an adequate therapeutic effect. CDCA is therefore less prone to causing major side effects. That is why it is the treatment of choice for the majority of clinical experts. While the most common side effect of CDCA therapy is diarrhea, it has also shown certain hepatotoxic effects and can cause liver enzyme elevation [12]. However, none of these side effects have manifested in our patients.

CDCA has a direct effect on gene expression in hepatic macrophages. It obtains its main metabolic effect by downregulating the hepatic synthesis of CYP7A1, an enzyme functioning as the rate-limiting step in the synthesis of bile acids by the classical pathway (through 7α-Hydroxycholesterol), which has been proven both in mice and humans [13]. The downregulation of CYP7A1 has also been shown to lower the risk of atherosclerosis in mice fed with a Western diet [14].

In CTX, the activity of hydroxymethylglutaryl-coenzyme A (HMG-CoA) is increased as a result of insufficient bile acid levels. This, in turn, increases the relative synthesis of cholesterol and upregulates the synthesis of low-density lipoprotein receptors, resulting in increased peripheral cholesterol depositions with normal serum cholesterol levels [15,16]. Combining CDCA with HMG-CoA reductase inhibitors suppresses the production of cholesterol, thus inhibiting xanthoma production. Combination therapy with LDL apheresis, CDCA and HMG-CoA reductase inhibitors can be an effective regimen, as it has been proven to lead to a faster decrease in serum cholestanol levels than standard CDCA treatment [17]. 

Another possible alternative mode of treatment could be gene therapy. In 2021, a study (Lumbreras S. et al.) exploring this option on CTX mice was published. An adeno-associated virus vector was used to supplement the *CYP27A1* gene in hepatocytes. The treatment led to the successful restoration of bile acid metabolism. The authors concluded that gene therapy could be a feasible treatment option in human patients in the future [18].

*CYP27A1* variants have been associated in the literature with other conditions, mainly of malignant nature. A study (Cho H. et al.) established that a regular expression of *CYP27A1* is imperative for the function of natural limonoid products such as harrpernoid D. These compounds have selective antiproliferative activity against *BRAF* and *NRAS*-mutation-dependent melanoma [19]. Two studies (Liang Z. et al., Zhang X. et al.) have concluded that the *CYP27A1*, as a result of its role in cholesterol homeostasis, inhibits the proliferation and migration of renal cell carcinoma cells [20,21]. A third study (Liang Z. et al.) emphasizes its importance when it comes to inhibiting the proliferation and migration of urinary bladder carcinoma cells [22]. To investigate the effects of *CYP27A1* gene deletion (loss of enzyme function) in tumor tissue on median overall survival time, we created an in silico analysis, as described in our previous bioinformatics study [23]. The cBioPortal platform revealed a shallow deletion of *CYP27A1* in 32.28% (501 cases) of bladder urothelial carcinoma, 14.98% (426 cases) of breast invasive ductal carcinoma, 7.23% (110 cases) of renal clear cell carcinoma, 6.43% (233 cases) of prostate adenocarcinoma and 2.04% (18 cases) of cutaneous melanoma. Patients with *CYP27A1* deletion had a significantly lower median overall survival time (101.07 (95% CI, 86.83–118.76)) compared to those without *CYP27A1* deletion (122.17 (95% CI, 115.93–128.53)) (*p* < 0.001). These findings show the importance of *CYP27A1* variants in the context of malignant diseases initiation and prognosis. It is therefore worth raising the question of whether continuous oncological screening should be indicated in CTX patients.

Metabolites and ratios of metabolites in newborn dried blood spot tests can be used to detect the disease in early childhood CTX screening [24]. 7α,12α-dihydroxy-4-cholesten-3-one, β-cholestane-3α,7α,12α,25-tetrol glucuronide (GlcA-tetrol)/tauro-chenodeoxycholic acid (t-CDCA) ratio and GlcA-tetrol are considered effective markers in CTX screening [25,26]. When a proband is diagnosed with CTX, it is advised to offer genetic counseling to the partner and offspring of the proband, especially in areas where a founder effect related to CTX can be found, such as in Israel, Morocco and in partners of Druze heritage, where there is a risk that they may also be heterozygous for a *CYP27A1* mutation [6]. It is helpful to provide pre-test and post-test counseling, as this provides information and emotional support to families and helps them in family planning [27]. Preimplantation and prenatal genetic testing are options if pathogenic mutations have been found in one of the parents [28]. Parents who are both known carriers of *CYP27A1* mutations are advised to undergo in vitro fertilization with preimplantation genetic diagnosis to exclude implantation of embryos with biallelic mutations in *CYP27A1* [6]. 

## 5. Conclusions

CTX leads to the accumulation of cholesterol metabolites in neuronal and muscular tissue, resulting in cognitive and motor developmental disorders in early childhood and the progressive degradation of motor capabilities in childhood or early adulthood. In patients with CTX, it is of great importance to make a diagnosis early and begin treatment so that the progression of the disease can be halted and even reversed. However, this case is an excellent example of why clinical genetic testing is imperative in adults when there is suspicion of a possible metabolic or genetic pathology that was not diagnosed in early childhood.

## Figures and Tables

**Figure 1 biomedicines-11-01430-f001:**
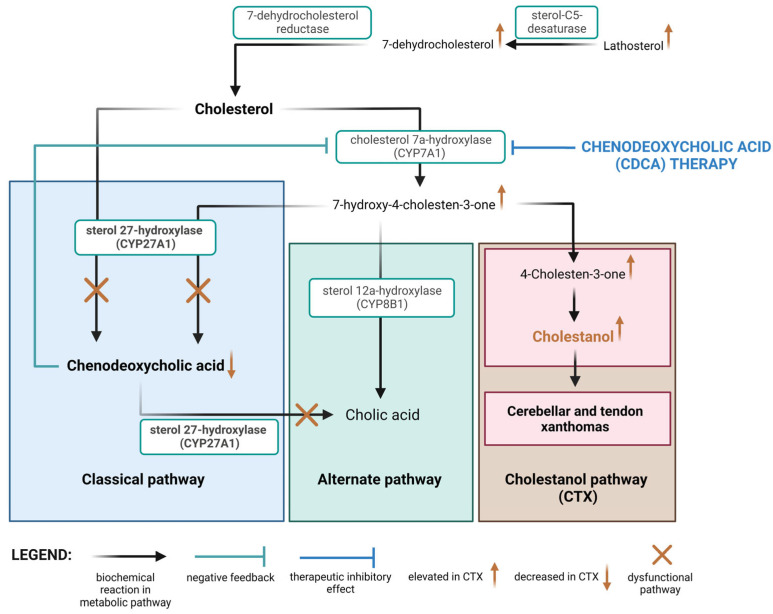
The molecular pathogenesis of CTX. The image shows the normally dominant metabolic pathways of cholesterol, whose activity is mainly regulated by CYP27A1. The absence of this enzyme leads to a significant increase in activity in the cholestanol pathway, as well as an increase in the upstream cholesterol metabolites. CDCA therapy inhibits CYP7A1 and, by extension, halts the cholestanol pathway and accumulation of cholestanol metabolites in the CNS. Created with BioRender.com.

**Figure 2 biomedicines-11-01430-f002:**
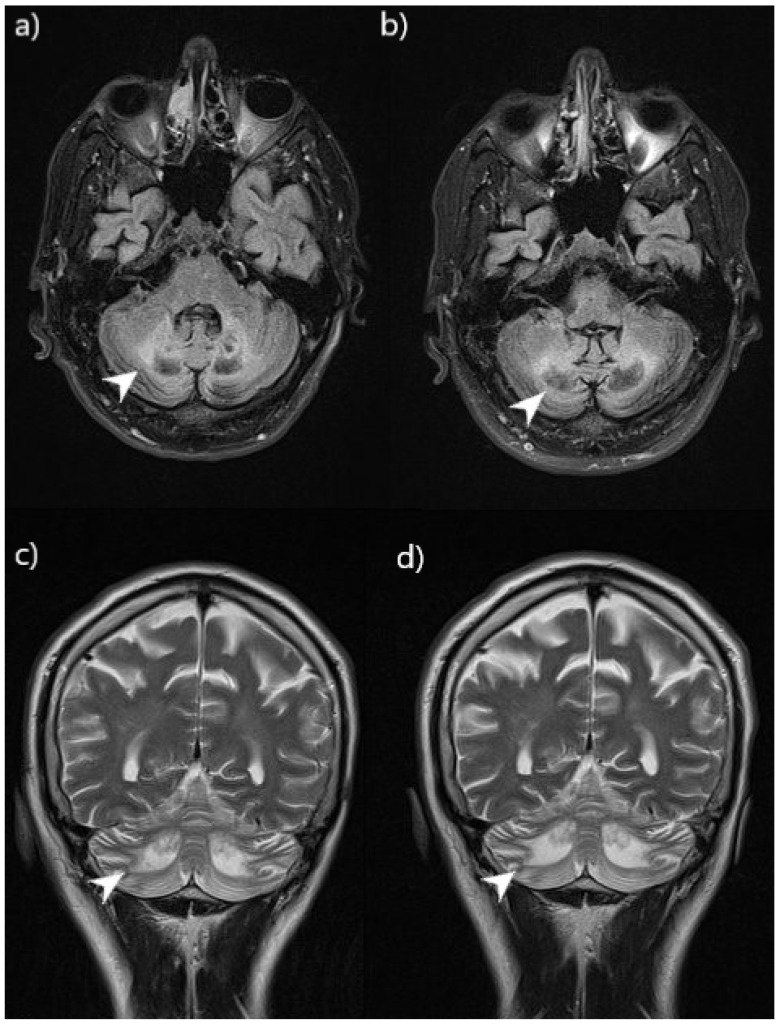
Comparison of CNS lesions on MRI (patient 1) before and after 9 months of CDCA therapy. Brain MRI showed bilaterally symmetrical hyperintensity in the cerebellar white matter and the dentate nucleus (arrowhead). Images (**a**,**c**) were taken before therapy and images (**b**,**d**) after. Images (**a**,**b**) were taken in the axial view, FLAIR sequence. Images (**c**,**d**) were taken in the coronal view, T2W sequence. The images show that there has been no progression of lesion growth.

**Figure 3 biomedicines-11-01430-f003:**
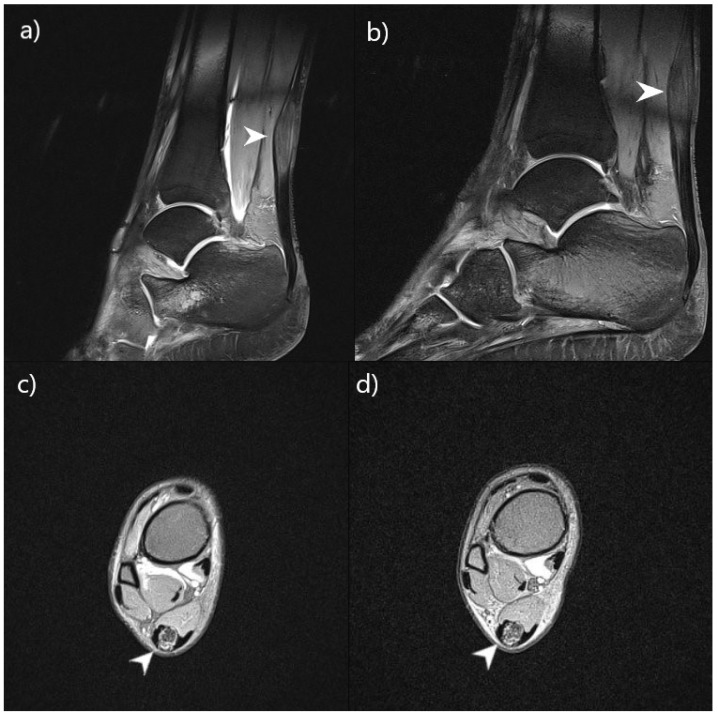
Comparison of lower leg lesions on MRI (patient 2) before and after 9 months of CDCA therapy. Ankle MRI showed focal fusiform thickening of both Achilles tendons, which corresponds to xanthoma accumulations in cerebrotendinous xanthomatosis (arrowhead). Images (**a**,**c**) were taken before therapy and images (**b**,**d**) after. Images (**a**,**b**) were taken in the sagittal view, true FISP sequence. Images (**c**,**d**) were taken in the axial view, PD FS sequence. The images show that there has been no progression of lesion growth.

**Table 1 biomedicines-11-01430-t001:** Phenotype characteristics of the patients and serum levels of cholesterol and its metabolites throughout treatment.

Time since Treatment Initiation	Before Treatment		3 Months		6 Months	
Characteristic	Patient 1	Patient 2	Patient 1	Patient 2	Patient 1	Patient 2
Epilepsy	+	−	+	−	+	−
Bilateral early onset cataracts	+	+	NA	NA	NA	NA
Thoracic kyphosis	+	+	+	+	+	+
Ataxia	+	+	Improved	Improved	Significantly improved	Significantly improved
Paraparesis	+	+	Improved	Improved	Significantly improved	Significantly improved
Dysarthria	+	+	Improved	Improved	Improved	Improved
Poor sphincter control	−	+	−	Improved	−	Improved
Diarrhea	+	+	Improved	Improved	Significantly improved	Significantly improved
Central nervous system abnormalities on MRI	+	+	/	/	Without progression	Without progression
Achilles tendon abnormalities on MRI	+	+	/	/	Without progression	Without progression
Cholestanol (<20.0 µmol/L)	139.6 (H)	140.1 (H)	42.5 (H)	51.2 (H)	10.1 (N)	8.8 (N)
Total Serum cholesterol (2.0–5.0 mmol/L)	4.5 (N)	3.8 (N)	6.5 (H)	6.4 (H)	/	/
Lathosterol (<10.0 µmol/L)	17.5 (H)	21.6 (H)	3.5 (N)	4.6 (N)	/	/
7-Dehydrocholesterol (<5.0 µmol/L)	71.1 (H)	58.3 (H)	2.5 (N)	2.3 (N)	/	/
ALT (12–48 U/L)	16 (N)	12 (N)	22 (N)	31 (N)	25 (N)	48 (N)
AST (11–38 U/L)	18 (N)	19 (N)	20 (N)	27 (N)	22 (N)	29 (N)
GGT (11–55 U/L)	14 (N)	10 (N)	20 (N)	/	19 (N)	16 (N)

H—higher than normal range; N—within normal range; (...)—reference interval; MRI—magnetic resonance imaging; +—exhibits symptom; −—does not exhibit symptom; /—not measured; NA—not applicable; ALT—alanine aminotransferase; AST—aspartate aminotransferase; GGT—gamma glutaryltransferase.

**Table 2 biomedicines-11-01430-t002:** Discovered genetic variants.

Gene	Variant	Zygosity	Variant Classification
PATIENT 1			
** *CYP27A1* **	**c.1184+1G>A (donor splice site)**	**Homozygous**	**Pathogenic**
*AGL*	c.1028G>A (p.Arg343Gln)	Heterozygous	VUS
*AMPD1*	c.133C>T (p.Gln45 *)	Heterozygous	VUS
*CASQ1*	c.280-3_280-2del (splice site)	Heterozygous	VUS
*CAV3*	c.449A>T (p.Glu150Val)	Heterozygous	VUS
*COL12A1*	c.4049C>A (p.Pro1350His)	Heterozygous	VUS
*ITGA7*	c.1601G>A (p.Arg534Gln)	Heterozygous	VUS
*SYNE2*	c.6328C>T (p.Pro2110Ser)	Heterozygous	VUS
PATIENT 2			
** *CYP27A1* **	**c.1184+1G>A (donor splice site)**	**Homozygous**	**Pathogenic**
*AMPD1*	c.133C>T (p.Gln45 *)	Heterozygous	VUS
*COL12A1*	c.4049C>A (p.Pro1350His)	Heterozygous	VUS

* VUS—variant of uncertain significance.

## Data Availability

The data presented in this study are available on request from the corresponding author.

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
