# Peer review of "Successful Treatment of a Rare Cholesterol Homeostasis Disorder Due to CYP27A1 Gene Mutation with Chenodeoxycholic Acid Therapy"

_biomedicines, 2023, doi:10.3390/biomedicines11051430_

Round 1

Reviewer 1 Report

The authors present two new cases of cerebrotendinous xanthomatosis diagnosed at 36-40 years of age, and characterize their response to the standard treatment (chenodeoxycholic acid, CDCA). The manuscript is well-organized and contributes the clinical experience about the response to different disease manifestations upon CDCA treatment. The discussion includes updated information about diagnostic/screening parameters, novel therapies, and provides an interesting comment about a potential increase in the oncogenic risk of patients.

These are some suggestions to improve the quality of the manuscript.

1.       Figure 1. Please provide a figure legend and a better representation of the biochemical pathways. Include the classical and alternative bile acid synthesis pathways, indicating the involvement of CYP27A1 in both of them.

2.       Indicate the age of onset of cataracts in patient 1.

3.       Table 1. The outcome of CDCA treatment on speech alteration in Patient 2 should be indicated (instead of N/A).

4.       The main effect of CDCA in the metabolic pathway consists of the inhibition of CYP7A1 upregulation. The addition of HMG-CoA inhibitors to the CDCA treatment is controversial and not widely implemented.

Please be very cautions when suggesting that other treatments such as cholic acid and ursodeoxycholic acid are acceptable alternatives to CDCA. Discontinuation of CDCA supply can be catastrophic to the patients. CDCA is the only treatment accepted by the majority of clinical experts. Substitution of CDCA by ursodeoxycholic acid, for instance, has been reported to cause significant patient deterioration (Brike et al., Images in Metabolic Medicine, 2021).

Author Response

Response to Reviewer 1 Comments

Reviewer 1 commented that the manuscript is well-organized and contributes clinical experience regarding the response to different disease manifestations upon CDCA treatment. He suggested a few minor modifications to improve the manuscript, which we made.

Point 1: Figure 1. Please provide a figure legend and a better representation of the biochemical pathways. Include the classical and alternative bile acid synthesis pathways, indicating the involvement of CYP27A1 in both of them.

Response 1: As you kindly requested, we have modified Figure 1. and have added a comprehensive figure legend at the bottom of the image. Both pathways are now shown in the figure and the involvement of CYP27A1 is shown in both of them.

Point 2: Indicate the age of onset of cataracts in patient 1.

Response 2: Thank you for your comment. Based on the patient's medical records, the bilateral cataracts were first detected during an ophthalmological examination at the age of 31 and were subsequently surgically treated. Based on your suggestion we have indicated this data in L127-130.

Point 3: Table 1. The outcome of CDCA treatment on speech alteration in Patient 2 should be indicated (instead of N/A).

Response 3: Thank you for your suggestion. As in the primary version of Table 1., speech alteration was not mentioned as a parameter in Table 1., we have exchanged Developmental delay in the onset of speech with the parameter “Dysarthria” in Table 1. and indicated improvement after CDCA treatment, as you suggested.

Point 4: The main effect of CDCA in the metabolic pathway consists of the inhibition of CYP7A1 upregulation. The addition of HMG-CoA inhibitors to the CDCA treatment is controversial and not widely implemented.

Response 4: We have taken your feedback into consideration and revised the manuscript accordingly. The revised version of the manuscript now includes the statement that CDCA obtains its metabolic effects by down-regulating cholesterol hydroxylation by CYP7A1 (L266-271). We emphasize the importance of treating patients with CDCA, for it is the only treatment option accepted by the majority of clinical experts (L260-262), while HGM-CoA inhibitors and gene therapy are indicated as possible alternatives and additional modes of treatment which are not widely implemented.

Reviewer 2 Report

I think it is a great article, well written and well explained even for a non-clinical researcher.  During review, the following concerns arose:

1.     In figure 1, it appears that 7a-hydroxy-4-cholesten-3-one can be converted to cholesterol, which can be converted to the sterol intermediates upstream in the synthesis pathway. What happens is that in the absence of the end product, enzymes are overexpressed in an attempt to compensate for the lack of the end product, causing us to see higher levels of the sterol intermediates. The red arrows in reverse in the cholesterol synthesis give rise to confusion, where the reader might think that the product gives rise to the above sterols, which is not the case. They should find another way of expressing what they want to say.

2.     They should indicate in the materials and methods which method they use to quantify cholestanol, cholesterol, lathosterol, 7-dehydrocholesterol.

3.     When table 1 shows cholesterol levels, is that data total cholesterol or free cholesterol?

4.     What are these patients' plasma total cholesterol, LDL-cholesterol and HDL-cholesterol levels?

5.     Do liver enzyme levels change before and after treatment?

Author Response

Response to Reviewer 2 Comments

Reviewer 2 commented that the study is well-designed, and the experimental approaches and the interpretation of the data are appropriate. He suggested a few minor modifications to improve the manuscript which we obliged:

Point 1: In figure 1, it appears that 7a-hydroxy-4-cholesten-3-one can be converted to cholesterol, which can be converted to the sterol intermediates upstream in the synthesis pathway. What happens is that in the absence of the end product, enzymes are overexpressed in an attempt to compensate for the lack of the end product, causing us to see higher levels of the sterol intermediates. The red arrows in reverse in the cholesterol synthesis give rise to confusion, where the reader might think that the product gives rise to the above sterols, which is not the case. They should find another way of expressing what they want to say.

Response 1: Thank you for your valuable feedback. The red arrows in reverse have been removed to avoid confusion, and we have marked which metabolites are increased in CTX so the idea is conveyed more clearly in the revised version of Figure 1.

Point 2: They should indicate in the materials and methods which method they use to quantify cholestanol, cholesterol, lathosterol, 7-dehydrocholesterol.

Response 2: As you have kindly requested, we have implemented this data in the Materials and Methods section, subsection Clinical Examination and Treatment (L94) mentioning that cholestanol, cholesterol, lathosterol, 7-dehydrocholesterol were quantified using gas chromatography and mass spectrometry.

Point 3: When table 1 shows cholesterol levels, is that data total cholesterol or free cholesterol?

Response 3: Thank you for your comment. The data in Table 1. is shown as total blood cholesterol. Therefore, we have changed the data title in Table 1. to “Total serum cholesterol”.

Point 4: What are these patients' plasma total cholesterol, LDL-cholesterol and HDL-cholesterol levels?

Response 4: To answer your question, we have included the total serum cholesterol, HDL, and LDL levels of both patients six months after treatment initiation in the "The Effect of Chenodeoxycholic Acid Treatment" portion of the Results section of the manuscript (L218-223), as was suggested.

Point 5: Do liver enzyme levels change before and after treatment?

Response 5: In response to your inquiry, we have updated Table 1 to include liver enzyme levels as a parameter tested before treatment initiation, three and six months after treatment initiation. We found that liver enzyme levels slightly increased but were all within normal levels three and six months after treatment initiation.

Reviewer 3 Report

This manuscript is well written and all sections are quite comprehensive. The authors have done a nice work in writing this article.

Major point

The main limitation of this study is the lack of clinical significance. It is just a case report of two patients with a rare disease. I do not see any advantage for clinicians and for researchers with the publication of this case report. Indeed, the clinical presentation of these two patients, their diagnostic results as well as their response to CDCA therapy does not add anything new to the literature. Everything is rather confirmatory of what is well known in the field.

Author Response

Response to Reviewer 3 Comments

Reviewer 3 commented that our manuscript is well written and all sections are quite comprehensive. However, he had some concerns regarding the clinical significance of our study, which we discussed below.

Point 1. The main limitation of this study is the lack of clinical significance. It is just a case report of two patients with a rare disease. I do not see any advantage for clinicians and for researchers with the publication of this case report. Indeed, the clinical presentation of these two patients, their diagnostic results as well as their response to CDCA therapy does not add anything new to the literature. Everything is rather confirmatory of what is well known in the field.

Response 1: Thank you for your review. We would like to address your concerns regarding the clinical significance of our study. Firstly, despite numerous reported cases of cerebrotendinous xanthomatosis (CTX), our study highlights that this disease is often not diagnosed early enough due to variations in clinical presentation and age of symptom onset. Our report presents cases of patients who were misdiagnosed for almost three decades with a diagnosis of Congenital cataracts with facial dysmorphism and neuropathy (OMIM: 604168) before receiving a correct diagnosis of CTX. Therefore, even with a wealth of knowledge about CTX, cases still exist where the disease is not detected early enough. Unfortunately, in these cases, therapy is less effective compared to treatment initiated before the age of 25. Nevertheless, our patients achieved excellent results despite their age, with significant improvements in speech, complete resolution of ataxia, and marked progress in motor function, in contrast to previously published literature that suggests insignificant clinical improvement with treatment at a later age. Thus, in addition to laboratory parameters (now within the reference range), our study provides insights into significant clinical improvement and effectiveness of CDCA therapy in patients treated in their third decade. Unusually, Patient 2 (younger brother) exhibited more severe symptoms than Patient 1 (older brother), despite being siblings and sharing the same genetic mutation, which highlights the influence of environmental factors (such as diet, daily activities, and natural environment) in the pathogenesis of CTX. Moreover, our patients did not have visible xanthomas that were observable solely through targeted MRI. Additionally, the loss of CYP27A1 function has been associated with an increased risk of developing malignant tumors in numerous studies, which warrants further investigation. Therefore, we used the publicly available cBioPortal database to investigate the significance of CYP27A1 loss of function in different types of tumors, analyzing a total of 19,823 samples. We also discussed the implications of this finding on the prognosis and follow-up of patients with CTX. Overall, we believe that the combination of our molecular and clinical approach to this disease highlights the importance of early diagnosis and proper patient care, which can lead to a favorable outcome.

Furthermore, our study provides updated information on multiple treatment options for CTX, which may be of interest to clinicians and researchers. It provides clear and concise information about the challenges in diagnosing CTX, the effectiveness of CDCA therapy, and the potential association between CTX and an increased risk of oncogenesis. We believe that our modified and improved manuscript effectively addresses the concerns raised about the clinical significance of the study. We hope that our findings can contribute to the advancement of CTX diagnosis and treatment, ultimately improving patient outcomes.

Round 2

Reviewer 1 Report

1. Cholesterol levels are usually not elevated in CTX patients. Please correct figure 1.

2. CDCA does not halt the metabolism of cholesterol. Please correct this sentence in the introduction.

Author Response

Response to Reviewer 1 Comments

Reviewer 1 suggested a few minor modifications to improve the manuscript, which we made.

Point 1: Cholesterol levels are usually not elevated in CTX patients. Please correct Figure 1.

Response 1: Thank you for your comment. We corrected Figure 1, as was suggested.

Point 2: CDCA does not halt the metabolism of cholesterol. Please correct this sentence in the introduction.

Response 2: We corrected this sentence in the introduction (L78-81), as you suggested.

We would like to thank reviewer 1 for all the suggestions and comments for improving our manuscript.

Reviewer 2 Report

The authors have answered all my questions satisfactorily, improving the manuscript.

Author Response

Response to Reviewer 2 Comments

Reviewer 2 commented that we answered all his questions satisfactorily and improved the manuscript.

Response: We would like to thank reviewer 2 for all the suggestions and comments for improving our manuscript.

Reviewer 3 Report

Although I appreaciate the work done by the authors, the main limitation of this study still exists. There is a lack of clinical significance. It is just a case report of two patients with a rare disease. I do not see any advantage for clinicians and for researchers with the publication of this case report. Indeed, the clinical presentation of these two patients, their diagnostic results as well as their response to CDCA therapy does not add anything new to the literature. Everything is rather confirmatory of what is well known in the field. Overall, I do not think that this article should be published in BIOMEDICINES and I suggest to target another journal. The Editorial Office may help you to find another proper journal.

Author Response

Response to Reviewer 3 Comments

Reviewer 3 commented that he appreciate our work, but that there is a lack of clinical significance. Overall, he suggested targeting another journal.

Response: We greatly appreciate the feedback provided by reviewer 3. However, we would like to emphasize that his comment appears to be in contrast with the opinions expressed by other reviewers. Other reviewers have acknowledged the value of our article in providing clinical insights on the effects of CDCA treatment on different disease manifestations. They have also acknowledged the value of the updated information included in our discussion on diagnostic and screening parameters, novel therapies, and the potential increase in oncogenic risk for patients.

Moreover, it is important to note that our manuscript is well-aligned with the scope of this special issue, which is centered on the crucial topic of "Lipid and Cholesterol Metabolism in Health and Disease: A Focus on the Cross-Talk between Peripheral Tissues and Central Nervous System". As such, we believe that transferring our manuscript to another journal would be inappropriate.

Once again, we appreciate the feedback provided by reviewer 3 and we hope our revised version is significantly improved.